# An evaluation of microphysics in a numerical model using Doppler velocity measured by ground-based radar for application to the EarthCARE satellite

Woosub Roh[1], Masaki Satoh[1], Yuichiro Hagihara[2], Hiroaki Horie[2], Yuichi Ohno[2], and Takuji Kubota[3]

[1]Atmosphere and Ocean Research Institute, The University of Tokyo, Kashiwa, Chiba 277-8564, Japan
[2]National Institute of Information and Communications Technology, Koganei, Japan
[3]Earth Observation Research Center, Japan Aerospace Exploration Agency, Tsukuba, Ibaraki, Japan

*Correspondence to*: Woosub Roh (ws-roh@aori.u-tokyo.ac.jp)

**Abstract.** The Cloud Profiling Radar (CPR) of the Earth Cloud, Aerosol and Radiation Explorer (EarthCARE) has a new capability to observe the Doppler velocity related to the vertical air motion of the terminal velocity of hydrometeors. The new observation from space will be used to evaluate and improve the model. Before the launch of EarthCARE, we need to develop a methodology for using the CPR data for model evaluations. In this study, we evaluated simulated data by a stretched version of the global non-hydrostatic model over Japan with a ground-based CPR using an instrument design similar to the EarthCARE CPR. We chose two cases with different precipitation events in September 2019 using two cloud microphysics schemes. We introduced the categorization method for evaluating microphysics using Doppler velocity. The results show that the liquid and solid phases of hydrometeors are divided in Doppler velocity, and the model's terminal velocities of rain, snow, and graupel categories can be evaluated with the observation. The results also show that the choice of microphysics scheme has a more significant impact than the dependence on precipitation cases. We discussed the application of the EarthCARE-like simulation results using a satellite simulator.

## 1 Introduction

Satellite data have been used to evaluate and improve clouds and precipitation of global circulation models (GCMs). Recently, global storm resolving models (GSRMs, Satoh et al. 2019; Stevens et al. 2019) have been used to produce more detailed simulations of mesoscale convective systems with km-scale horizontal mesh, which is at a much finer resolution than typical GCMs. GSRMs are expected to reduce the uncertainty of GCMs due to a cumulus parameterization. GSRMs implement cloud microphysics schemes to achieve realistic simulations of clouds and precipitation by considering the micro-physical processes of hydrometeors such as nucleation, coalescence, and precipitation in the model. Most of the along-track sampling of the active satellite sensors is less than 5 km, which is comparable to the horizontal resolution of GSRMs. Therefore, it is possible to directly compare the satellite data and the results of GSRMs without any subgrid assumption. Several studies have evaluated and improved microphysics using satellite active sensor data (e.g., Roh and Satoh 2014; Roh et al. 2017; Ikuta et al. 2021).

One of the innovative satellite projects is the Earth Cloud, Aerosol and Radiation Explorer (EarthCARE, Illingworth et al. 2015; Wehr et al. 2023) satellite, which is a joint mission between the European Space Agency

(ESA) and the Japanese Aerospace Exploration Agency (JAXA). EarthCARE has multiple passive and active sensors in the same spacecraft to investigate clouds, aerosol, precipitation and associated radiation budgets. It has Cloud Profiling Radar (CPR), ATmospheric LIDar (ATLID), Multi Spectral Imager (MSI) and Broad Band

Radiometer (BBR). The CPR of EarthCARE has the Doppler capability to obtain information on the terminal velocity of hydrometeors and vertical air motions. The multiple sensors of EarthCARE will provide additional information that will enhance our understanding of the interaction between clouds and aerosols.

New observations, such as the Doppler velocity from EarthCARE, will provide new insights into the evaluation and improvement of GSRMs. Before launching the satellite, it is important to understand how to use

the information of the Doppler velocity to evaluate GSRMs. We use the observation of the Doppler velocity by a cloud radar installed on the ground and investigate the methodology of evaluating the GSRMs using a sensor simulator for the Doppler velocity. We use the W-band Doppler cloud radar at Koganei-shi in Japan. This radar was installed with a similar setting to the EarthCARE CPR. We use the ground radar to understand how to use the observation by the EarthCARE CPR before the launch.

The ground remote sensing observation data in Japan are highly concentrated in metropolitan areas due to disaster prevention. The ULTra-sIte for Measuring Atmosphere of Tokyo metropolitan Environment (ULTIMATE, Satoh et al. 2022) project plans to use the extensive observational data from Tokyo metropolitan area together with satellite observations to evaluate and improve the cloud microphysics schemes of GSRMs. In this project, we used various types of radars, including the dual polarization Doppler C-band radars (Satoh et al.

2022, Ikuta et al. 2022), X-band polarimetric radars, Ka-band cloud radars, and wind profilers. The W-band Doppler cloud radar at Koganei-shi is one of the radars used in the ULTIMATE project.

A satellite simulator can generate EarthCARE-like signals before the satellite launch. Roh et al. (2023) produced the EarthCARE-like radiances by a GSRM and the Joint-Simulator for Satellite Sensors (Joint-Simulator; Hashino et al., 2013; Roh et al., 2020). The dataset is referred to as the EarthCARE synthetic data.

These data were used to study the EarthCARE satellite retrieval algorithm (Hagihara et al., 2021; Wang et al., 2022; Hagihara et al., 2023). Hagihara et al. (2021) investigated the characteristics of the Doppler velocity using the EarthCARE synthetic data produced by Roh et al. (2023). They investigated the unfolding correction and the impact of the increase in the horizontal sampling to reduce the random errors of the Doppler velocity.

Several studies relate the ground Doppler velocity to cloud microphysics. Han et al. (2013) evaluated the four

different microphysics in the Weather Research and Forecasting (WRF) model using radar reflectivity and Doppler velocity from S-band radar. Burns et al. (2016) investigated marine stratiform clouds with radar reflectivity and Doppler velocity using ground-based W-band radar and the CPR simulator (Kollias et al. 2014). However, these studies did not focus on quantitative analysis of hydrometeors for microphysics.

One of the motivations of this study is to evaluate and compare the vertical distribution of hydrometeors of

GSRMs using the same observational criteria. According to Roh et al. (2021), the horizontal distribution of outgoing longwave radiation of GSRMs is similar, but the simulated vertical distributions of hydrometeors of GSRMs are very different in the intercomparison data (Stevens et al. 2019). Each model used its own assumptions about the size distribution and terminal velocity of hydrometeors. We believe that the Doppler velocity is one of the criteria for understanding and constraining the vertical distributions between GSRMs using

observations.

In this study, we develop a new evaluation method for a cloud microphysics scheme using the vertical profile of the Doppler velocity. We use the ground observational data. The methodology can be applied to the EarthCARE observation. We evaluate two types of cloud microphysical schemes using this method. We investigate the EarthCARE-like simulations using the Joint-Simulator and discuss the results from different instrument settings with random errors.

The observational data and the settings of the simulation data are described in section 2. An evaluation method and results are presented in section 3. The application of the EarthCARE-like simulation data is discussed in Section 4. A summary is given in section 5.

## 2 Data and methodology

We used the Nonhydorstatic Icosahedral Atmospheric Model (NICAM; Satoh et al., 2014) as a GSRM. We followed the approach by Roh and Satoh (2014) to use NICAM as a regional model by transforming the grid to focus on the region of interest with high resolution (the stretched NICAM; Tomita 2008a). We conducted NICAM simulations using a G-Level 10 (GL10) horizontal resolution with the stretch-factor of 100 (the ratio between the maximum and minimum grid intervals), where the minimum grid interval is approximately 800 m. We evaluated two microphysics schemes in NICAM: Single-moment Water 6-categories (Tomita 2008b) with modifications by Roh and Satoh (2014) (hereafter referred to NSW6) and the NICAM Double-moment Water 6-categories (Seiki and Nakajima 2014, hereafter referred to NDW6). The NICAM simulations were initialized using the National Centers for Environmental Prediction (NECP) data with a one-degree resolution for wind, temperature, relative humidity and geopotential data. The sea surface temperature was fixed.

We simulated two cases of rain events in September 2019. The first case (case 1) is the tropical cyclone (TC) Faxai. The second is a weak frontal system (case 2). In case 1 the integration and analysis time was from 00 UTC on 8 September to 00 UTC on 9 September 2019. In case 2 the integration and analysis time was from 00 UTC on 20 September to 00 UTC on 21 September 2019.

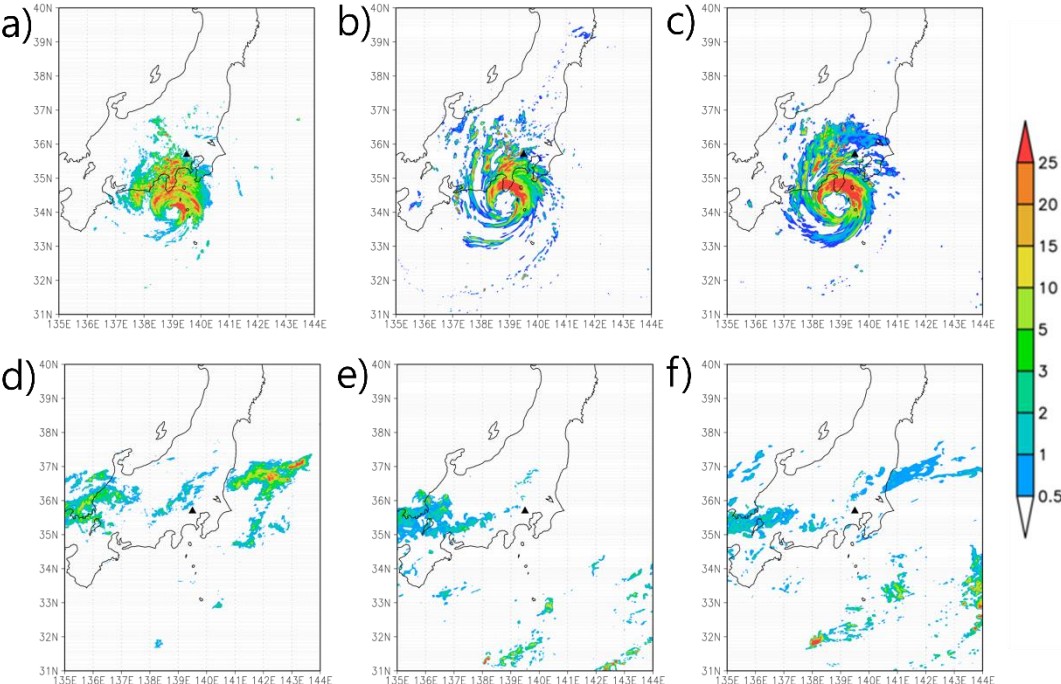

**Figure 1:** The horizontal distributions of the precipitation for the observation [radar/rain gauge–analyzed precipitation] (left), NSW6 (center), and NDW6 (right) simulations for case 1 (top) and case 2 (bottom). The black triangle indicates the location of the CPR in NICT. The contour is the precipitation rate [mm hr$^{-1}$].


We used the High-sensitivity Ground-based Super Polarimetric Ice-crystal Detection and Explication Radar (HG-SPIDER, Horie et al. 2000) at the National Institute of Information and Communications Technology located at (35.7°N, 139.5°E). This HG-SPIDER is a vertically pointing radar and performs similarly to the EarthCARE CPR with Doppler capability. The vertical sampling resolution of HG-SPIDER is 75 m, and the

observation range is 11.475 km. The time interval is less than a second, and we used one-minute integrated data for the analysis. For case 1, the data are only available for 12 hours.

The sensor simulator used for the evaluation of HG-SPIDER was the EarthCARE Active Sensor Simulator (EASE, Okamoto et al. 2007, 2008; Nishizawa et al. 2008). We make the same assumptions about the size distributions of hydrometeors both for the NICAM simulation and the Joint-Simulator. For cloud ice and cloud

water of NSW6, size distributions are not explicitly assumed in NICAM. For these categories, we used the mono size distributions for the effective radius of cloud ice as 40 μm and the effective radius of cloud water as 8 μm in the Joint-Simulator.

Figure 1 shows the horizontal distributions of precipitation for case 1 and case 2. Case 1 has heavy precipitation in the Tokyo metropolitan area associated with the rain bands of TC Faxai (Fig. 1a). Case 2 shows

weak precipitation in the analysis area with scattered precipitation distribution (Fig. 1d). The two NICAM simulations capture the rainbands similarly to the observation for case 1 (Fig. 1b, c) and the frontal system for case 2 (Fig. 1e, f). However, NSW6 missed the precipitation system over the Pacific Ocean near 36° and 37° latitudes. For case 1, NICAM simulated the structure and the track of TC Faxai similarly to the observation, but the simulated TC Faxai moved faster than the observation (not shown here). For the statistical analysis, we

define the analysis domain as the Japan area in Fig. 1.

## 3 Results

### 3.1 Observation by the ground CPR

Figure 2 shows the Contoured Frequency by Altitude Diagrams (CFADs) of the radar reflectivity and the Doppler velocity observed by the ground CPR. It was created using the 1.25 dBZ bins and the 0.25 m/s bins at each height increment (75 m). For case 1, the radar reflectivity rapidly decreases from the ground up to a 4 km altitude, and the maximum radar reflectivity is less than 5 dBZ. The upward decrease in the radar reflectivity is due to the strong wet attenuation from rain and the wet attenuation on the antenna's radome. Attenuation is not dominant in case 2 because of the weaker precipitation than in case 1. In case 1, the vertical profiles of the radar reflectivity are scattered, and no specific pattern is evident. In case 2, the reflectivity increases from the upper layer to the lower layer due to the growth of ice particles above the melting layer, and the melting layer can be seen more clearly than in case 1. In general, the ground observation is not free to rain attenuation, which significantly affects the reduction of radar reflectivity, especially for precipitation cases. Hereafter, we focus on the Doppler velocity evaluations.

Figure 2c and d show the CFAD of the Doppler velocity. The benefit of the Doppler velocity is that it is free from attenuation. The Doppler velocity is the sum of the terminal velocity and the vertical air motion. If we assume that the vertical air motion is relatively small, the Doppler velocity is related to the terminal velocity of the hydrometeors. A negative Doppler velocity means hydrometeors falling toward the ground, whereas a positive Doppler velocity means upward motion. There are two different modes above and below the melting layer. These two modes are the fast terminal velocity of rain below 5 km and the slow terminal velocity of ice hydrometeors above 5 km. We believe that since case 1 has heavy precipitation, the riming process which produces mostly large rimed ice such as graupel and hail, and it has a Doppler velocity less than -2m/s is dominant. There are frequencies above 2 m s$^{-1}$ in case 1 below 5 km altitude. These higher values of the Doppler velocity are related to the aliasing effect with rain having a terminal velocity greater than 5.4 m/s. One must be careful about the aliasing effect for analysis of upward motion

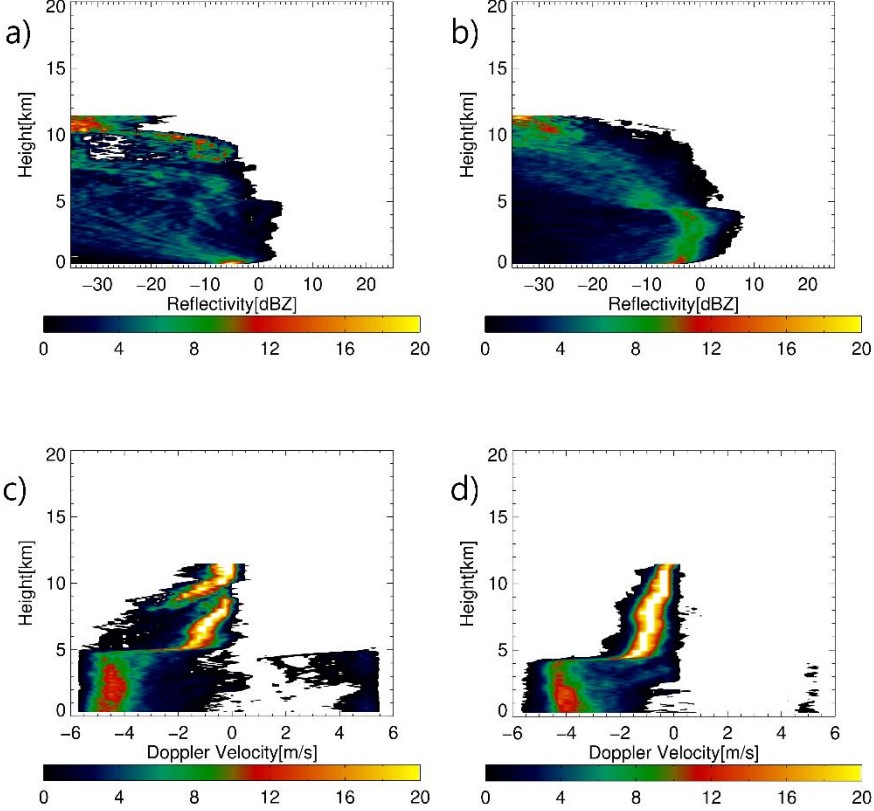

**Figure 2:** The CFADs of the radar reflectivities (top) and Doppler velocities (bottom) for case 1 (left) and case 2 (right). The unit of the contour is the normalized frequency at each height. The bin widths for the height and the Doppler velocity are 75m and 0.25 m/s..

### 3.2 Categorization of hydrometeors

We introduce a categorization method for hydrometeors using the probability frequency of Doppler velocity in height (see figure 3). The figure was constructed with 0.25 m/s bins and 75 m bins (see figure 3). We identified five regimes: (1) graupel/hail, (2) cloud ice (CI)/snow, (3) rain, (4) cloud water (CW)/drizzle and (5) upward motion using thresholds of Doppler velocity with -2 m/s, 0 m/s, and height with 5 km. We use the unfolding method based on Hagihara et al. (2021) to reduce the aliasing effect. They applied the unfolding method for the Doppler velocity above 3 m/s:

$$V_{unfoled} = V_{folded} - 2 * V_{max} \text{ for } V_{folded} > 3\text{m/s},$$

where $V_{max}$ = 5.4 m/s is for this instrument.

According to the Glossary of Meteorology of the American Meteorological Society, the diameter of a drizzle is less than 0.5 mm, and the terminal velocity is 2.068 m/s with 0.5 mm at the surface based on Foote and Toit 1969. Mosimann 1995 investigated the degree of snow crystal riming using vertical Doppler radar. He found that the degree of riming is proportional to the Doppler velocity and that there is a large fraction of graupel with the Doppler velocity greater than 2 m/s (fig. 3 in Mosimann 1995). In this classification we did not consider the air density effect. This classification has uncertainty from vertical air motion and air density. We think the

impact of these two terms is not significant. This study does not aim for an accurate classification of hydrometeors but rather for a quantitative intercomparison of models on the same basis.

We have applied this method to the two cases and found the characteristics of the precipitation systems between the two cases. Note that the unfolding method is useful to reduce the aliasing effect in case 1. The rain fraction is dominant in case 1 (56.2%), and the CI/snow fraction is dominant in case 2 (39.0%). The proportion of graupel/hail is higher in case 1 (6.9%) than in case 2 (0.3%). The graupel/hail fraction is large in case 1, suggesting the importance of the riming process with convective rain. The upward motion fraction is higher in case 1 (5.5%) than in case 2 (1.0%), but the total fraction is less than 6%. We can summarize that rain dominates in case 1, while CI/snow dominates in case 2. Using this categorizing method, we can quantify the dominant hydrometeors of the precipitation systems.

The thermodynamic transition height is 5 km. The maximum height of the ice-to-liquid transition is slightly lower than the melting layer (Klaassen 1988). The melting layer is lower in case 2 than in case 1. The melting layer depends on the seasonal variance at mid-latitude. To apply the categorization in more general cases, including the tropical area and the mid-latitudes, we will extend it using the threshold of temperature at the freezing point 0°C instead of the height 5 km.

195

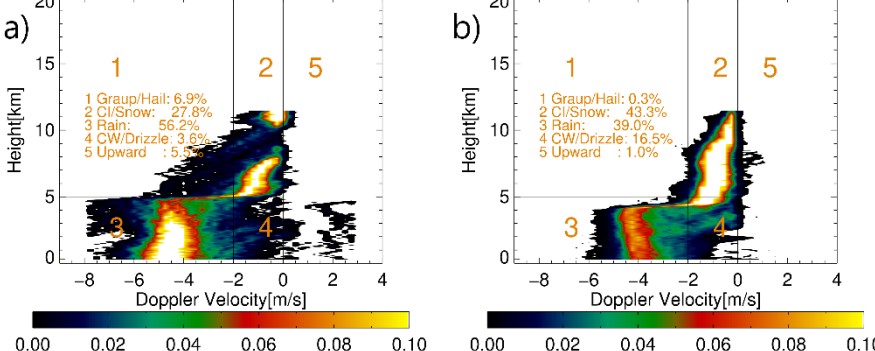

**Figure 3:** The categorizations of the hydrometeors using the joint histogram between Doppler velocity and height for case 1 and case 2.

200

### 3.3 Evaluation results

We assumed the contribution of vertical air velocity to Doppler velocity is relatively smaller than the terminal velocity of hydrometeors. If the absolute vertical air velocity is larger than the terminal velocity of hydrometeors, the categorization method produces a large bias and makes the results unreliable. We investigated this issue using the NICAM simulation data. Figure 4 shows the cumulative probability density functions (PDFs) of the absolute vertical air velocity with radar echo larger than -40 dBZ with a 0.2 m/s bin for the calculation. We

found that the frequency of absolute vertical velocity above 0.2 m/s is less than 0.2 %, and the simulated PDF of the Doppler velocity mostly depends on the cloud microphysics. NSW6 shows more contribution of the vertical air velocity on the Doppler velocity than NDW6. Case 1 has a higher fraction of the absolute vertical air velocity greater than 0.2 m/s. Because stronger convection is produced with the tropical cyclone, the vertical velocity affects the accuracy of this categorization in NSW6. Note these results are affected by the horizontal resolution of the model (e.g. Lebo and Morrison 2015). When we used the coarse resolution, the contribution of the vertical air motion was larger than the finer horizontal resolution (not shown).

215

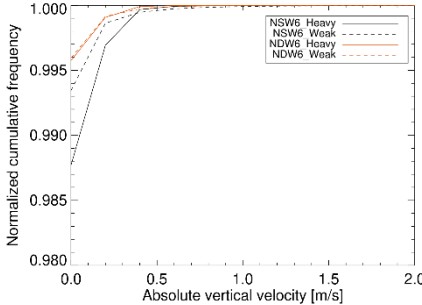

**Figure 4:** The cumulative PDF of the absolute vertical air velocity for the sampling data with larger than -40 dBZ in NICAM. NSW6_Heavy: case 1 with NSW6, NSW6_Weak: case 2 with NSW6, NDW6_Heavy: case 1 with NDW6, and NDW6_Weak: case 2 with NDW6.

220

We categorize CI/snow if the terminal velocity is less than 2 m/s, and graupel/hail if the terminal velocity is greater than 2 m/s. We used the same separation threshold between rain and cloud water or drizzle. The categorization results depend highly on the cloud microphysical schemes. NSW6 and NDW6 use different terminal velocity assumptions for each ice hydrometeor (Fig. 5a, b). Using 2 m/s as the threshold, NSW6 has a clear separation of the categorization between CI/snow and graupel/hail compared to NDW6. The terminal velocity of rain is similar between NSW6 and NDW6 (Fig. 5c). The drizzle with a diameter less than 0.5 mm is slower than 2 m/s of terminal velocity in both NSW6 and NDW6. NSW6 shows the greater terminal velocity of raindrops with less than 0.5 mm diameter. The definitions of hydrometeors are different. The evaluation of the same criterion is more important than the direct comparison of hydrometeors. We can understand the effect of terminal velocity using Doppler velocity in precipitation systems.

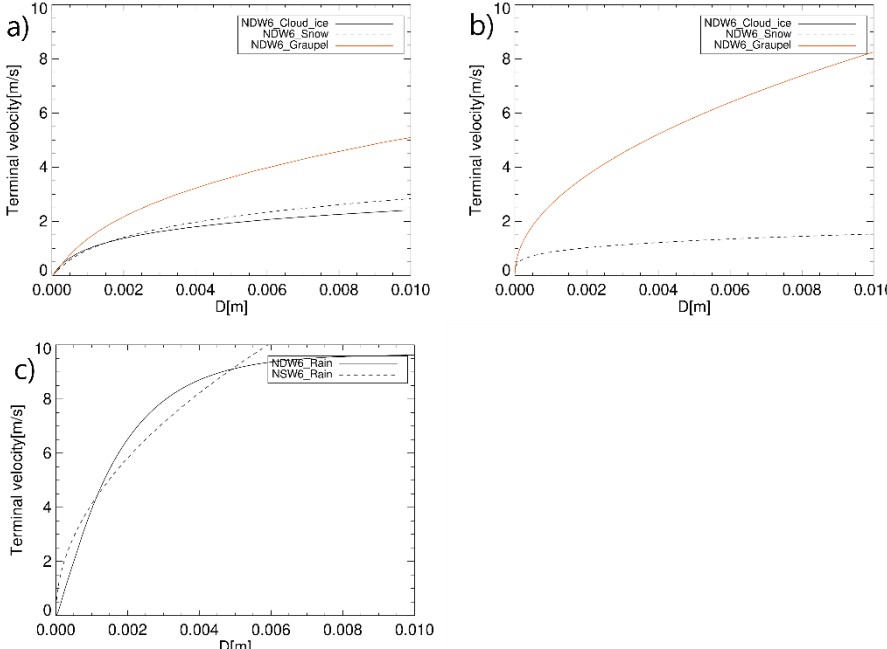

**Figure 5:** Terminal velocities against the diameter of ice hydrometeors in NSW6 (a) and NDW6 (b), and the diameter of rain (c).

In Figure 6, we evaluated the NICAM results using this method. We simulated the Doppler velocities for the entire analysis domain for the large data sampling. Both microphysical schemes reproduced the two observed branches: fast terminal velocity of rain and slow terminal velocity of ice hydrometeors. The two microphysics simulations show similar case dependencies. NSW6 shows a lower graupel/hail fraction than the observation and NDW6. There is a higher fraction of graupel/hail and rain in case 1 than in case 2 in NSW6 and NDW6. There is a low fraction of CW/drizzle in case 1 (0.6%) than in case 2 (3.6%) in NSW6. The choice of microphysics has a more significant effect than the case dependencies.

There are more fractions of ice hydrometeors in NDW6 than in the observation and NSW6 in case 1. It means that there are 63% of ice hydrometeors (graupel/hail and CI/snow) in the simulations in NDW6. The observation is less than 35% of ice hydrometeors. We can expect a large fraction of ice hydrometeors to affect the radiation in the simulations.

The fraction of CW/drizzle is underestimated in both simulations. NDW6 performs better than NSW6. However, the two simulations do not reproduce the fraction of CW/drizzle in case 2. One of the reasons is that the horizontal resolution is too coarse to simulate the low clouds in case 2.

NDW6 shows the growth of snow from cloud ice more clearly than NSW6. The transition height is 9 km in case 1 and 7 km in case 2 in NDW6. NDW6 overestimates the graupel/hail regime associated with large snow or ice crystals. This result indicates that the terminal velocity of the snow is overestimated compared to the observation.

The melting layer is reproduced as the difference in terminal velocity between ice and liquid hydrometeors in NSW6. However, NDW6 does not show this contrast due to the high graupel/hail fraction.

We checked the impact of the vertical air motion on the joint histogram with calculated Doppler velocity without vertical air motions using the Joint-Simulator (Fig. 7). The impact of vertical air motion does not significantly affect the results, as shown in Fig. 4. The impact of vertical air motion is mostly within 2% and the difference is most pronounced in case 1 with strong convection. In NSW6, the variance of frequency tends to be smaller.

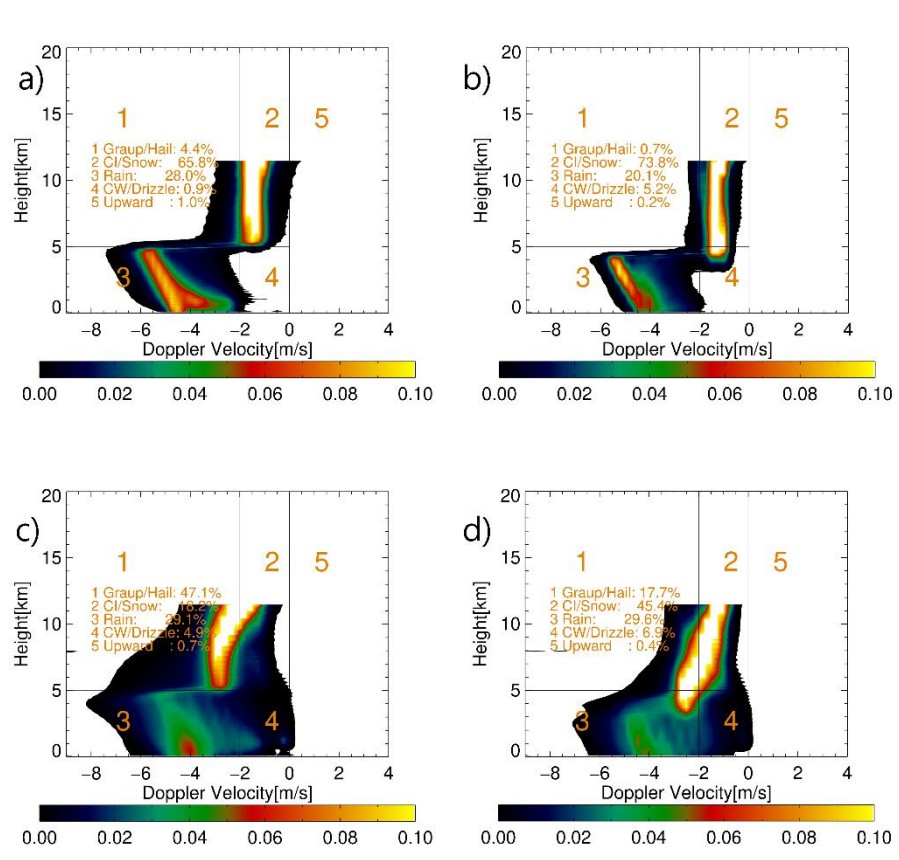

**Figure 6:** The categorizations of the hydrometeors in NICAM simulations for NSW6 (top) and NDW6 (bottom) in case 1 (left) and case 2 (right).

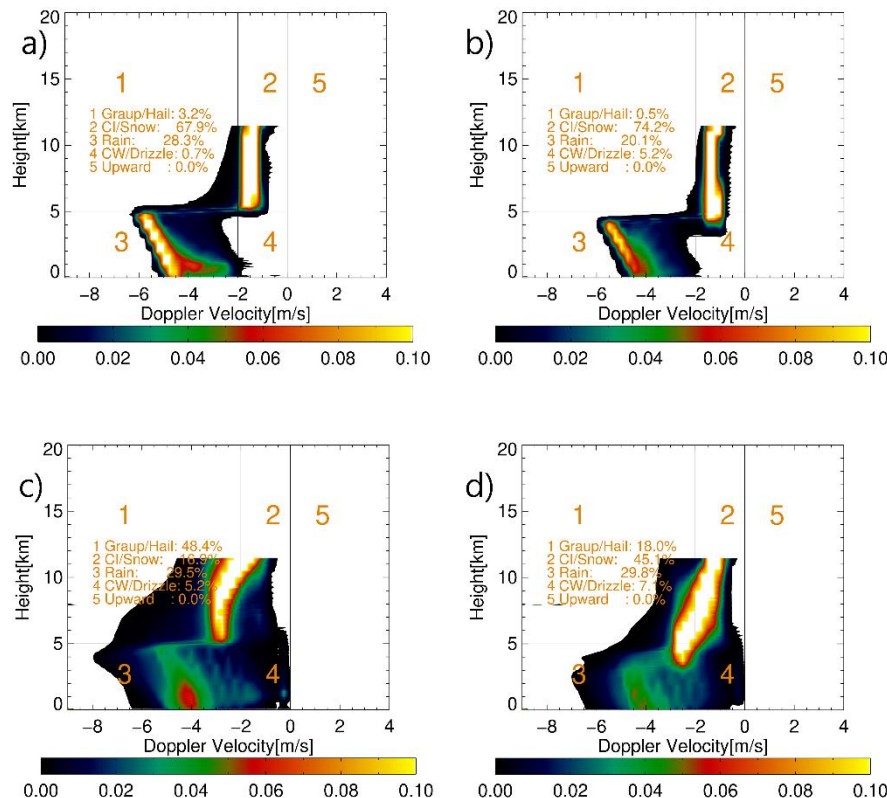

**Figure 7:** The same as Figure 6 but for only calculations of Doppler velocity without vertical air motion.

## 4 Discussion on the EarthCARE-like data

### 4.1 Evaluation results of the EarthCARE-like data

We simulated the EarthCARE-like data using NICAM and the Joint-Simulator. We increased the observation
window to 20 km and changed the vertical resolution to 99.9308 m based on the setting of the EarthCARE CPR.
We set the minimum detectable radar reflectivity to -36 dBZ. Increasing the observation window increased the
sampling of ice hydrometeors. It decreased the sampling of liquid hydrometeors and upward motion (Fig. 7).
Note that we changed the sign of the EarthCARE Doppler velocity so that it has the same direction as the
Doppler velocity for the ground observations, and we changed the bin size of the height from 75 m to 99.9308 m.
The autoconversion process from cloud ice to snow in NSW6 was shown at an altitude of 14 km, which was not
visible in the ground-based simulation data (Fig. 8 a, b). The accuracy of the Doppler velocity is related to the
signal-to-noise ratio (SNR). Therefore we analyzed the data with a radar reflectivity greater than -15 dBZ based
on Hagihara et al. (2021). In this case, we found an increase in the fraction of hail/graupel and rain regimes than
all sampling data (Fig. 9). The fraction of CI/snow is reduced compared to the other regime. The results are
consistent with the ground observation data regarding case dependency related to the riming process. The
characteristics of the microphysics also show a similar dependence related to the fraction of graupel/hail and the

fraction of CW/drizzle between NSW6 and NDW6. However, if the radar reflectivity threshold increases, the cloud echo related to small cloud ice and cloud water vanishes.

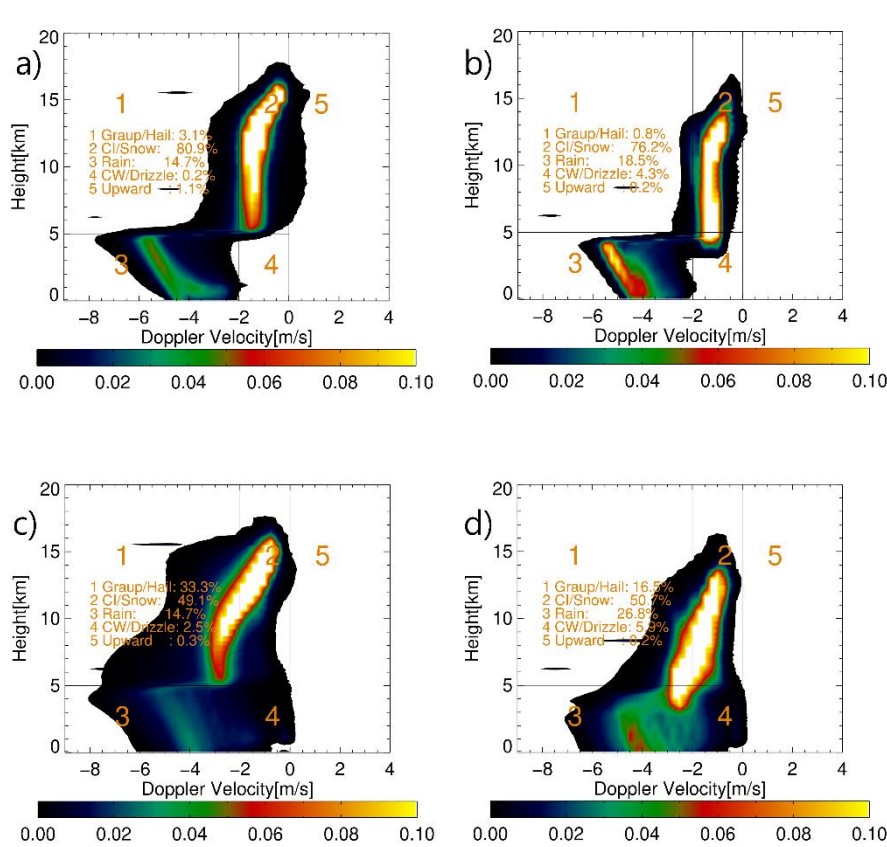


**Figure 8:** The categorizations of the hydrometeors based on the EarthCARE-like simulations for NSW6 (top) and NDW6 (bottom) in case 1 (left) and case 2 (right).

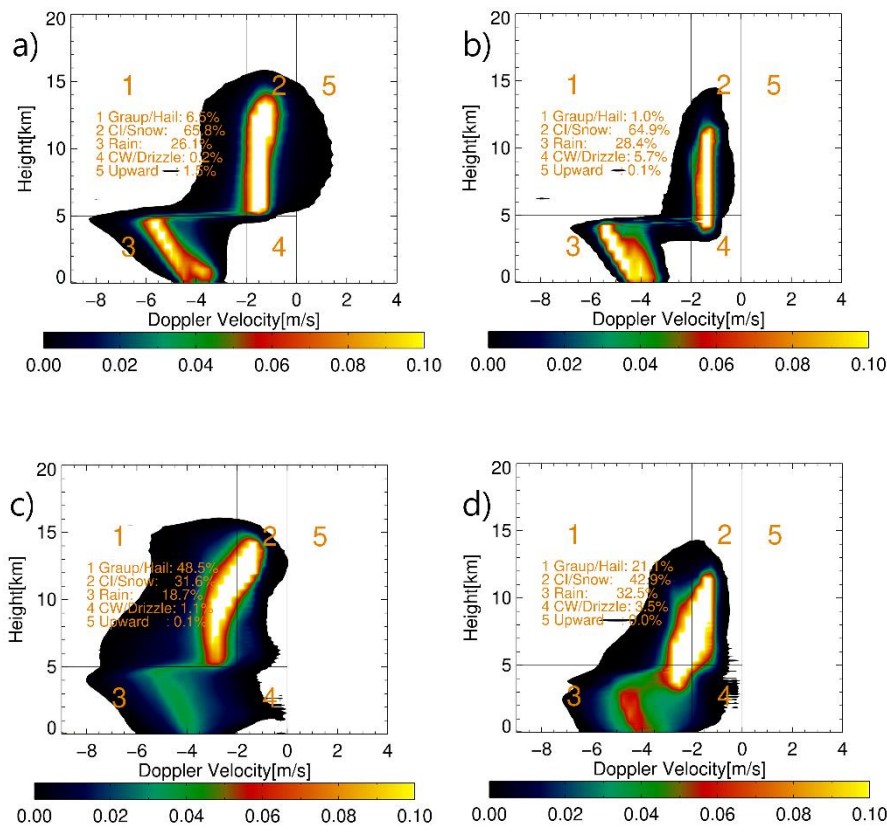


**Figure 9:** The same as Figure 8 but for only larger radar reflectivity than -15 dBZ.

## 4.2 Sensitivity tests of observation windows

The Joint-Simulator can simulate the EarthCARE CPR with possible random errors based on the investigation by Hagihara et al. (2021). Three modes of the observation window are considered for the EarthCARE CPR; the high, middle, and low modes observe up to the 20, 18, and 16 km altitudes at the top of the observation, respectively. The high and low modes will be used depending on latitudes: low mode (−1 to 16 km) at latitudes of 60°–90° and high mode (−1 to 20 km) at latitudes of 0°–60° (Hagihara et al., 2021). The high mode has a

higher observation window but lower Pulse Repetition Frequencies (PRFs) than the low mode. The low mode has better accuracy of Doppler velocity observation by the higher PRFs than the high mode. Figure 10 shows examples of a cross-section of Doppler velocities in case 1, comparing the Doppler velocity with no errors, high mode errors, and low mode errors. It shows that the low mode captures Doppler velocity similarly to the true Doppler velocity, whereas the high mode hardly observes the true magnitude of Doppler velocity. Although the

low mode shows better quality of the Doppler velocity observation, the observation limited to below 16 km is insufficient for the low latitude areas in the tropics.

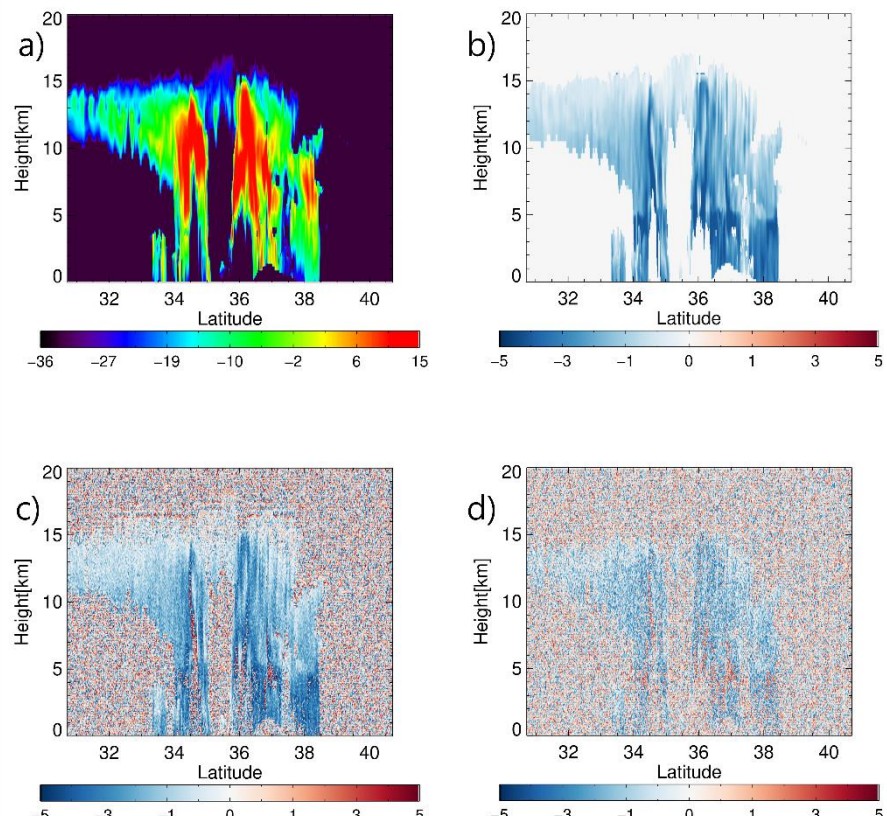

**Figure 10:** The cross-sections of simulated radar reflectivity (a), Doppler velocity without random error (b), Doppler velocity with the low mode (c), and Doppler velocity with the high mode for the area of Typhoon Faxai with the NICAM-NSW6 in case 1. The contour units are dBZ for radar reflectivity (a) and m/s for Doppler velocity (b, c, d).

We evaluated the Doppler velocities of the EarthCARE-like simulations with NSW6 and NDW6 with possible random errors based on the two observation window modes. First, we investigated the low mode results (Fig. 11). The random errors cause broadening of the variance of Doppler velocity. The results are consistent with the results with no random errors in Fig. 9. The difference between case 1 and 2 is overall similar to the signals with no error shown in Fig. 9. We see more fraction of the graupel/hail regime and CW/drizzle in NDW6 than NSW6 (Fig. 9 and Table 2). However, there is an increase in the fraction of the graupel/hail and upward motion fraction from the broadening of variances of Doppler velocity. For the high mode, the high fractions for each regime are diverged (Fig. 12 and Table 3). It is hard to distinguish the characteristics of the microphysics in the high mode.

We found that random errors degraded the accuracy of the hydrometeor classification compared to the true values with no error consideration. However, the results of the low mode show similar patterns of microphysics and case dependency to the true values. In this study, we did not consider the integration effect on these results. The official product of Doppler velocity is 1 km and 10 km integrated Doppler velocity along the orbit of the EarthCARE. According to Hagihara et al. (2021), when we use a 10 km integration for cloud echoes above -15 dBZ of radar reflectivity in the high mode, the standard deviation was less than 0.5 m/s. When we use 10 km integration data with the high mode, we can get similar results to the results of the low mode. We checked the resolution dependence using NICAM simulations with a four times coarser horizontal grid; the characteristics of

the joint histogram are consistent with the higher resolution simulation results. We expect the 10 km integration data with the high mode to be very useful for the evaluation of GSRMs.

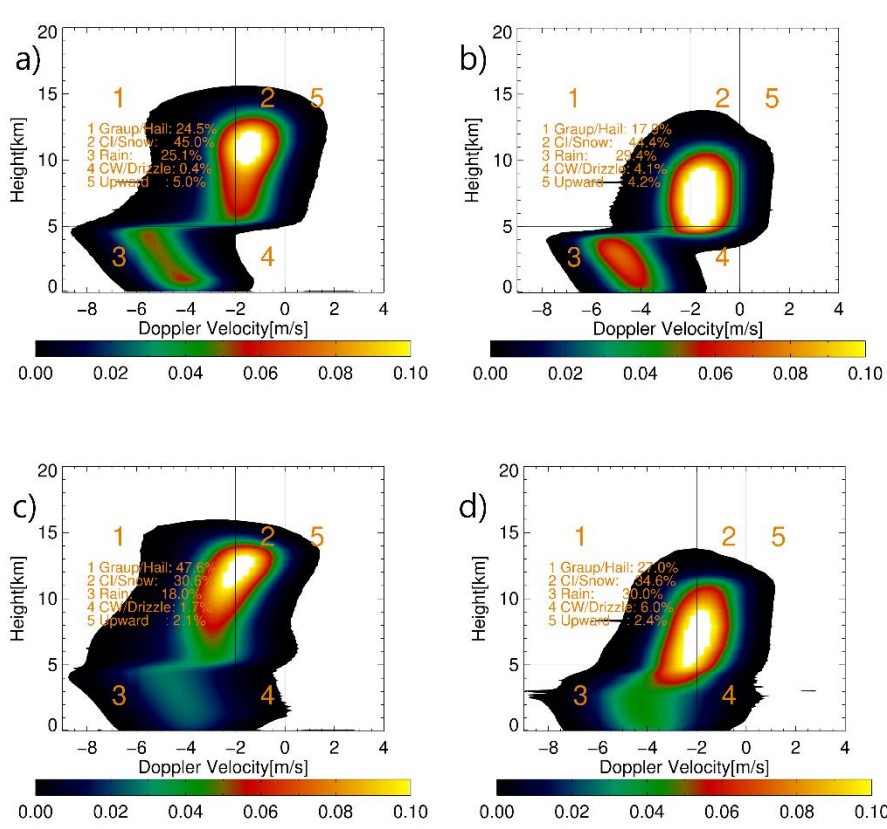

**Figure 11:** The categorizations of the hydrometeors based on the low mode of EarthCARE-like simulations for NSW6 (top) and NDW6 (bottom) in case 1 (left) and case 2 (right).


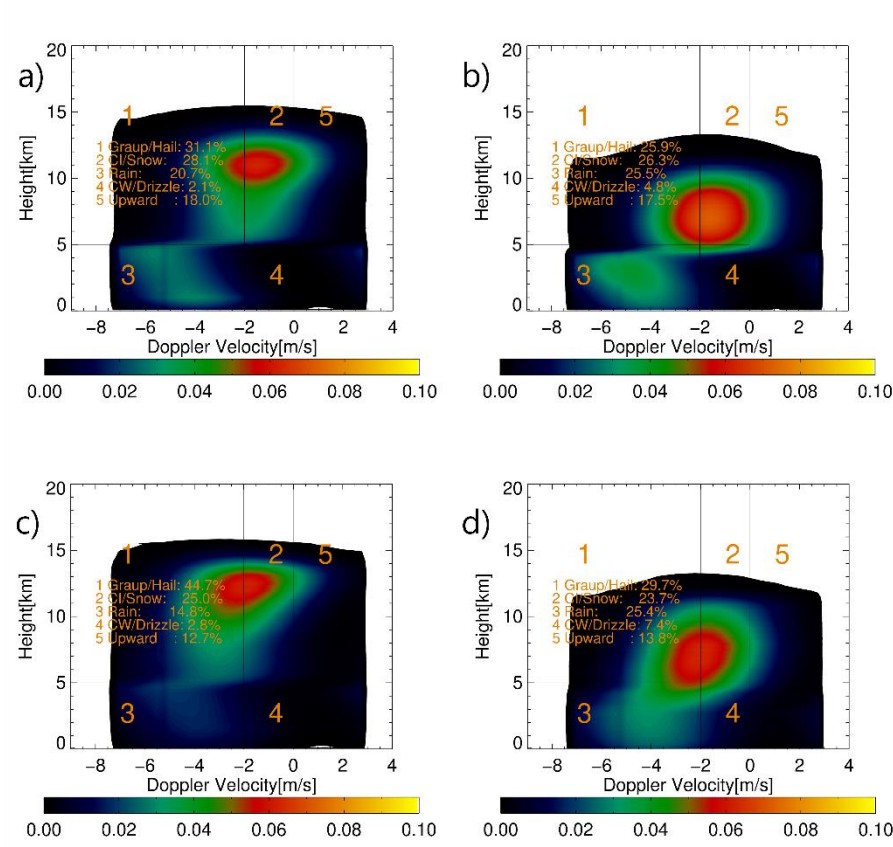

**Figure 12:** The categorizations of the hydrometeors based on the high mode of the EarthCARE-like simulations for NSW6 (top) and NDW6 (bottom) in case 1 (left) and case 2 (right).



**Table 1: The fraction of categorizations of the hydrometeors based on the EarthCARE-like simulations for only larger radar reflectivity than -15 dBZ from Fig. 9.**

|  | 1.Graupel/Hail | 2.Cloud Ice/Snow | 3.Rain | 4.Cloud Water/Drizzle | 5.Upward motion |
|---|---|---|---|---|---|
| NSW6 case1 | 6.5% | 65.8% | 26.1% | 0.2% | 1.5% |

| | 1.Graupel/Hail | 2.Cloud Ice/Snow | 3.Rain | 4.Cloud Water/Drizzle | 5.Upward motion |
|---|---|---|---|---|---|
| NSW6 case2 | 1.0% | 64.9% | 28.4% | 5.7% | 0.1% |
| NDW6 case1 | 48.5% | 31.6% | 18.7% | 1.1% | 0.1% |
| NDW6 case2 | 21.1% | 42.9% | 32.5% | 3.5% | 0.0% |

**Table 2: The fraction of categorizations of the hydrometeors based on the low mode of the EarthCARE-like simulations for only larger radar reflectivity than -15 dBZ from Fig. 11.**

| | 1.Graupel/Hail | 2.Cloud Ice/Snow | 3.Rain | 4.Cloud Water/Drizzle | 5.Upward motion |
|---|---|---|---|---|---|
| NSW6 case1 | 24.5% | 45.0% | 25.1% | 0.4% | 5.0% |
| NSW6 case2 | 17.9% | 44.4% | 29.4% | 4.1% | 4.2% |
| NDW6 case1 | 47.6% | 30.6% | 18.0% | 1.7% | 2.1% |
| NDW6 case2 | 27.0% | 34.6% | 30.0% | 6.0% | 2.4% |


**Table 3: The fraction of categorizations of the hydrometeors based on the high mode of the EarthCARE-like simulations for only larger radar reflectivity than -15 dBZ from Fig. 12.**

| | 1.Graupel/Hail | 2.Cloud Ice/Snow | 3.Rain | 4.Cloud Water/Drizzle | 5.Upward motion |
|---|---|---|---|---|---|
| NSW6 case1 | 31.1% | 28.1% | 20.7% | 2.1% | 18.0% |
| NSW6 case2 | 25.9% | 26.3% | 25.5% | 4.8% | 17.5% |
| NDW6 case1 | 44.7% | 25.0% | 14.8% | 2.8% | 12.7% |
| NDW6 case2 | 29.7% | 23.7% | 25.4% | 7.4% | 13.8% |

## 5 Summary

In this study, we developed a methodology for using the Cloud Profiling Radar (CPR) of Earth Cloud, Aerosol and Radiation Explorer (EarthCARE) before the launch of EarthCARE in 2024 for model evaluations. We analysed observation data by the ground-based CPR for two cases with different precipitation events in September 2019. By using the observed data, we compared simulated results for these cases by the stretched version of the global non-hydrostatic model, NICAM, with two different cloud microphysics schemes.

We introduced a categorization method of hydrometeors for analyzing Doppler velocity observed by CPR. This method is based on a joint histogram of Doppler velocity with respect to heights. We identified five regimes: (1) graupel/hail, (2) cloud ice (CI)/snow, (3) rain, (4) cloud water (CW)/drizzle and (5) upward motion. This method clarifies the contribution of the hydrometeors to the precipitation systems and characterize cloud microphysics. For the case of the tropical cyclone with heavy precipitation, the rain and graupel/hail fraction are more dominant than the weak precipitation case.

We applied the Joint-Simulator to the NICAM simulation data with two cloud microphysics schemes and analyzed the simulated Doppler velocity data using this categorizing method. These simulations produce a similar horizontal distribution of precipitation to the observation. The cloud microphysics schemes strongly impact the joint histogram of Doppler velocity in terms of heights, particularly for the heavy precipitation case. The double moment scheme reproduced a higher fraction of the graupel/hail regime than the observation and the single moment scheme.

The advantage of the use of Doppler velocity in the categorization of the hydrometeors is that Doppler velocities suffer less impact from the attenuation of rain and wet attenuation on an antenna. The ground CPR observation of the radar reflectivity for the precipitation case is limited because of wet attenuation on an antenna. The Doppler velocity of the ground observation is more reliable than the radar reflectivity. Doppler velocities are from the terminal velocity of the hydrometeors and vertical air motion. Analysis of the simulation results revealed that the main contribution to the Doppler velocity is the terminal velocity of hydrometeors. The terminal velocity has information about the density and thermodynamics phases of hydrometeors.

We expanded this evaluation method using the simulated Doppler velocities of the EarthCARE satellite. The results are consistent with the ground observation data. The maximum observation height of the EarthCARE CPR is higher than the ground observation. We tested about the threshold of radar reflectivity with -15 dBZ for the Doppler velocity. The results are consistent with the simulation data larger than -36 dBZ. However, there was an increase in the fraction of the ice hydrometeors and a decrease of the CW/drizzle because of the increase of the observation range and the threshold of the radar reflectivity.

We considered the observation windows and random errors associated with the Pulse Repetition Frequencies (PRFs). When we added the random error based on the observation window, the Doppler velocities diverged from the results without error. The 16 km observation window mode has the higher PRFs and reproduced consistent results similarly to the results without error. The differences between the two cloud microphysics schemes are apparent, such as the difference between the ground observation and the simulation with the 16 km observation window. In contrast, the 20 km observation window produces more random errors, and it was difficult to distinguish the different characteristics between two cloud microphysics. For the evaluation of cloud microphysics, the 16 km observation window is preferable, but higher clouds than the 16 km altitude would be

no longer omitted over the tropical region. Alternatively, if the product with the 10 km integration and the high

mode were used for model evaluation, we would expect to get the same consistent results as with the low mode.

This study did not account for the complexities of multiple scattering effects (Battaglia et al., 2011) and pointing uncertainties (Tanelli et al., 2005) in simulating Doppler velocities. These aspects are crucial for accurately assessing the Doppler velocity capabilities on the EarthCARE, including the impact of Pulse Repetition Frequencies (PRFs). Understanding these influences is challenging prior to the satellite's launch. We

will investigate these effects on the evaluation result in the future.

The merit of the CPR observation from space is to get better radar reflectivity for ice hydrometeors because of no attenuation from liquid hydrometeors than the ground observation data. The combination of the radar reflectivity and Doppler velocity has more information about size distribution and terminal velocity of ice hydrometeors. The EarthCARE has other three instruments. These instruments can detect the different

information related to cloud and precipitation systems.

In this study, the observation data are located in the middle latitudes. The melting layer changes between the two cases and the categorization between the ice and liquid hydrometeors has a bias because of the different melting layer heights. We will improve the transition height in the future for the middle latitudes.

After the launch of the EarthCARE satellite, the Doppler velocity can be available over the globe. The

Doppler velocity is more directly related to the terminal velocity of hydrometeors and characterizes cloud microphysics. To improve global storm-resolving models (GSRMs), the vertical distribution of hydrometeors must be more realistic by referring to available observations. The categorization method proposed in this study will quantify the hydrometeors simulated by GSRMs and lead to their improvement.

*Data availability*

The snapshot data of simulated radar reflectivity and Doppler velocity of 94 GHz Cloud Profiling Radar (CPR) are available from https://doi.org/10.5281/zenodo.10813626 (Roh et al., 2024). We made Fig. 10 using these data. The Joint-Simulator is available from https://www.eorc.jaxa.jp/theme/Joint-Simulator/userform/js_userform.html.


*Author contributions*

WR drafted the manuscript, and produced the NICAM simulation and simulated Doppler velocity using the Joint-Simulator. MS contributed the NICAM data and the manuscript. YH developed the random error model based on the observation window. HH and YO worked on the W-band observation in NICT. TK led the Joint-

Simulator development and provided feedback on the manuscript draft.

*Competing interests*

The authors declare that they have no conflicts of interest.

*Acknowledgements*

The authors thank members of the JAXA EarthCARE Science Team and the Joint-Simulator project. The authors also thank to ESA for providing the measured value of response functions of EarthCARE/MSI. The authors thank Dr. Toshi Matsui for providing the orbit/scan simulator. Computational resources were partly provided by the National Institute for Environmental Studies.

*Financial support*

    This work was supported by the EarthCARE satellite study commissioned by the Japan Aerospace Exploration
Agency. MS and WR were supported by a Grant-in-Aid for Scientific Research B (20H01967) and the Program for Promoting Technological Development of Transportation of the Ministry of Land, Infrastructure, Transport, and Tourism (MLIT).

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
