# Peer review of "An evaluation of microphysics in a numerical model using Doppler velocity measured by ground-based radar for application to the EarthCARE satellite"

_EGUsphere, 2023_

## Author Response (AR1)

We appreciate the reviewers' constructive comments. We have revised our manuscript accordingly, and we hope the reviewer will find the revisions satisfactory.

**Review of "An evaluation of microphysics in a numerical model using Doppler velocity measured by ground-based radar for application to the EarthCARE satellite" by Roh et al., submitted to Atmospheric Measurement Techniques (AMT)**

[Article#: amt-2023-1997]

**To Reviewer 1:**

Recommendation: Major revision

The manuscript deals with several aspects of how new EarthCare Doppler velocity (hereafter DV) measurements might address problems of hydrometeor identification using a variety of observations and simulations.

There are a number of interesting results in the manuscript, but ultimately I don't think the methodology or goals were explained well enough for me to have a clear understanding of what this study is really trying to communicate. For example, is the focus of this paper to define a DV-based hydrometeor classification system, or is that just a tool? I'm really unclear on the underlying "story" this work is telling. More clearly defining the goals of the study, and steps to reach that goal, will go a long way toward bringing this study into a publishable state.

The scope of the manuscript is within the main subject areas of AMT, specifically theoretical calculations of measurement simulations with detailed error analysis, including instrument simulations.

I suggest a major revision. I have no deep expertise in the measurements using each remote-sensing instrument. However, from such a perspective, the current manuscript needs to be improved for better readability and clearer points. In addition, the present descriptions of the data availability could be better because this manuscript aims to introduce and advance the use of the EarthCARE synthetic data by other engineers and researchers for retrieval algorithm development. I list major problems in the following section.

**Major comments:**

1. The manuscript is difficult to read because of English grammatical issues. A fairly extensive grammatical revision is required.

As an example, the first paragraph of the introduction would benefit from these changes:

(a) Remove the word "The" from the first sentence

(b) "Satellite" and "Global" should be lower case (first and second sentences)

(c) "The detailed process of hydrometeors" in fourth sentence does not make sense

The grammatical errors are too extensive to list. I would recommend that they enlist some help in the interest of ensuring that they are communicating their work effectively to their audience.

➔ The revised draft was checked by a researcher whose first language is English based on your comment.

2. The hydrometeor classification scheme (section 3.2) is abruptly stated, with no underlying justification offered. Where did these categories come from (specifically the apparently arbitrary absolute values that define the cutoffs)? How much uncertainty is present in these categories?

➔ We added the explanations based on some references. According to the Glossary of Meteorology of the American Meteorological Society, the diameter of a drizzle is less than 0.5 mm, and the terminal velocity is 2.068 m/s with 0.5 mm at the surface based on Foote and Toit 1969. Mosimann 1995 investigated the degree of snow crystal riming using vertical Doppler radar. He found that the degree of riming is proportional to the Doppler velocity and that there is a large fraction of graupel with the Doppler velocity greater than 2 m/s (fig. 3 in Mosimann 1995). In this classification, we did not consider the effect of air density. This classification has uncertainty from vertical air motion and air density. We think the impact of these two terms is not significant. This study does not aim for an accurate classification of hydrometeors but rather for a quantitative intercomparison of models on the same basis.

3. One major lingering question I have after reading the manuscript is as follows: For the NICAM simulations, the authors obviously know the distribution of hydrometers in the model. Why not compare their suggested classification scheme to what's actually present in the model? Maybe they did in fact do this, and I simply misunderstand the meaning of the figures. For example, in Figure 6(a), is 63.1% the fraction of cloud ice/snow derived from the DV-based classification scheme? Or is the cloud ice/snow fraction that was actually present in NICAM?

➔ We think the accuracy of the classification's names is not very important in this study. Microphysics scheme has different definitions of hydrometeors, their own terminal velocity, and size distributions. We think characteristics of vertical profiles of Doppler velocity in models related to terminal velocities of hydrometeors are more important. There are several uncertainties with this categorization. Even if it's cloud ice/snow, it's possible that there are mixtures of hydrometeors like small graupel. But we can understand that the average terminal velocity in that grid is high or low, and that's expected to have an impact on clouds and precipitation. We will investigate the impact of tuning of the Doppler velocity on radiation and large circulation in a global storm-resolving model (GSRM).

4. The authors later use fall-speed relationships from the model to at least somewhat clarify the hydrometeor classification system that was chosen. This leads to the larger point that the chronology of the "story" that is being told by this manuscript is somewhat muddled. The introduction should spell this out much more clearly, including the hierarchy of simulations that will be used.

➔ We added the motivation of this study about the evaluations of GSRMs using the Doppler velocity in detail in the introduction part: One of the motivations of this study is to evaluate and compare the vertical distribution of hydrometeors of GSRMs using the same observational criteria. According to Roh et al. 2021, the horizontal distribution of outgoing longwave radiation of GSRMs is similar, but the simulated vertical distributions of hydrometeors of GSRMs are very different in the intercomparison data (Stevens et al. 2019). Each model used its own assumptions about the size distribution and terminal velocity of hydrometeors. We believe that the Doppler velocity is one of the criteria for understanding and constraining the vertical distributions between GSRMs using observations.

5. The radar used for ground-based observations (HG-SPIDER) is polarametric, correct? Why not use the polarametric data to better quantify the actual hydrometeor classifications, and therefore enhance the classification system and associated discussion the physics?

➔ This radar (HG-SPIDER) is not a polarimetric radar.

6. The authors state that "We found that the frequency of absolute vertical velocity above 0.2 m/s is less than 2 %, and the simulated PDF of the Doppler velocity mostly depends on the cloud microphysics." I'm not convinced that the fact that absolute vertical velocities above 0.2 m/s are a small proportion of the total (Figure 4) means that air velocity can be neglected when translating Doppler velocity to particle fall speed. Since these results are from NICAM, why not just look at this relationship directly in the model? For example, heavy rain (although it's a small proportion of cases with reflectivity exceeding -40 dBZ) is more likely to occur in regions with significant vertical ascent.

➔ We investigated the impact of vertical air motion on the Doppler velocity using the Joint-Simulator. We removed the vertical air velocity for the calculation of Doppler velocity (Fig. 2). When we removed the vertical air velocity, the results were consistent with the control results (Fig. 1). However, the frequencies were concentrated in NSW6, and there was no fraction of the upward category. Most of the difference is less than 2% in the classifications. We added Fig. 2 and explanations in the revised draft.

[Figure]

Figure 1: The categorizations of the hydrometeors in NICAM simulations for NSW6 (top) and NDW6 (bottom) in case 1 (left) and case 2 (right).

[Figure]

Figure 2: The same as Figure 1 but for only calculations of Doppler velocity without vertical air motion.

7. What procedure is used to produce Figure 6, and how does it differ from the procedure used to produce Figure 7? Figure 6 apparently does not use the Joint simulator, so what does it use? Why does Figure 6 cut off at about 12 km when NICAM clearly produces results above 12 km?

The rather significant changes between Figures 6 and 7 are dealt with in a cursory way, but the changes are significant and what causes them need to be better explained.

➔ We used the Joint Simulator for Figure 6 and Figure 7. The differences between Fig. 6 and Fig. 7 are the setting of ground observation and the EarthCARE satellite. The observation range of the ground observation is up to 12 km, and the vertical resolution is different from EarthCARE. The CFADs of radar reflectivity are different because of the attenuation of rain. However, the results of Doppler velocity are very similar to ground observation. We expect there is an impact on data of Doppler velocity larger than -15 dBZ because of attenuation. Before we introduce the impact of random errors based on the observation window, we need to introduce the simulated Doppler velocity like EarthCARE.

**Specific comments:**

- Line 121: Do you mean that Doppler radar is free of attenuation?

➔ The radar reflectivity is attenuated. The Doppler velocity is not attenuated. However, the accuracy of Doppler velocity changes because of the attenuation.

- Line 182: Looks more like about 3 m/s, doesn't it?

➔ The terminal velocity of NDW6 is less than 2m/s, and the terminal velocity of NSW6 less than 3 m/s. I added a sentence like "NSW6 shows the faster terminal velocity of raindrops with less than 0.5 mm diameter.".

- Line 189: What is the "large data sampling"?

➔ The observation data is every 1 second. The model output data is every 1 hour. So we need to have a larger sampling of data for statistical analysis.

- Line 192: Check the 0.6% number.

➔ I checked the number.

- Line 215: Can you better define what is meant by "observation window"?

→ The observation window is different from the radar range. The observation window means a collected data range. The observation window depends on the PRFs, number of integration of pulse(M), and satellite altitude. The PRFs and M changes by the lookup table related to the satellite altitude. The observation window of CloudSat is 30 km (Tanelli et al. 2008).

- Line 221: Please briefly explain the -15 dBZ (in addition to the reference).

→ The errors of the Dopler velocity depend on the signal-to-noise (SNR). The lower SNR means the higher contribution of the signal noise to Doppler velocity. According to Hagihara et al. (2021), the standard deviation of random errors increases significantly when the radar reflectivity is less than −15 dBZ (SNR =6.2 dB).

- NDW6 acronym is not defined.

→We added the explanation about NDW6 like "the NICAM Double-moment Water 6-categories (Seiki and Nakajima 2014, hereafter referred to NDW6)".

- Figures 2 and 3 have no axes labels.

→ We added the axes labels.

- There is no reference to Figures 3 or 8 in the text.

→ We added the references.

- The inline text in Figures 6, 7, 8, and 10 is unreadable in some cases, since the text overlaps the contours. In Figure 3, the "3" and "4" are difficult to discern in places.

→ We improved all figures except Figure 1.

References

Foote, G. B., & Du Toit, P. S. 1969: Terminal velocity of raindrops aloft. *Journal of Applied Meteorology (1962-1982)*, 249-253.

Mosimann, L.: An improved method for determining the degree of snow crystal riming by vertical Doppler radar. *Atmos. Res., 37,* 305–323, doi:10.1016/0169-8095(94)00050-N, 1995.

Hagihara, Y., Ohno, Y., Horie, H., Roh, W., Satoh, M., Kubota, T., and Oki, R.: Assessments of Doppler velocity errors of EarthCARE cloud profiling radar using global cloud system resolving simulations: Effects of Doppler broadening and folding, IEEE Trans. Geosci. Remote Sens., 60, 1–9, https://doi.org/10.1109/TGRS.2021.3060828, 2021.

Kobayashi, S., Kumagai, H., & Kuroiwa, H.: A proposal of pulse-pair Doppler operation on a spaceborne cloud-profiling radar in the W band. *Journal of Atmospheric and Oceanic Technology*, *19*(9), 1294-1306, 2002.

Tanelli, S., Durden, S. L., Im, E., Pak, K. S., Reinke, D. G., Partain, P., ... & Marchand, R. T.: CloudSat's cloud profiling radar after two years in orbit: Performance, calibration, and processing. *IEEE Transactions on Geoscience and Remote Sensing*, *46*(11), 3560-3573, 2008.

**To Reviewer 2:**

Recommendation: Major revision

This study aims to use model simulations of two precipitation events by NICAM and a satellite simulator to investigate how spaceborne Doppler velocity (Vdop) measurement could be useful to evaluate microphysics, with an emphasis on categorization of hydrometeor types using thresholds in Doppler velocities. The topic is relevant to the satellite mission, EarthCARE. The language in the manuscript, , the justification of the scientific methods, and the presentation of the study needs to be significant improved before it can be considered for publication.

**Major comments:**

It is not clear why the authors chose to use threshold of the Vdop to categorize hydrometeor types. From a storm resolving model, all the hydrometeors are known. However, the authors chose not to use that information, but applying thresholds directly to the Vdop from a forward model (or satellite simulator). Why does this manuscript do it this way? Is it because the EarthCARE retrieval algorithm have such a component? Even if it is trying to mimic a component in the retrieval algorithms, storm resolving model could be better used to provide context for retrievals. For example, snow and graupel (other hydrometeors too) must be mixed in a large number of model grid points. How could the categorization of Vdop (such as those shown in Fig. 6) could be used to single out the mass of a certain hydrometeor type? Do you need to consider a ratio of mixing of hydrometeor types? Is that the retrieval would provide, or not?

The presentation of the study and figures need to be improved. I suggest adding a figure showing the observed reflectivity and Doppler velocities. as well as the simulated reflectivity and Vdop. Figure 1 is only the precipitation rate. The text in Figs. 3, 6, 7, 8, 10, 11 are not illegible. A table could be necessary for comparisons. The 2% in Fig. 4 is not accurate (should it be 0.3% for grid points with absolute air motion > 0.5 m/s?). The magnitude of the model-resolved vertical air motion and the model horizontal resolution also needs to be elaborated a bit more. Ignoring vertical air motion, directly relating Fall velocity to Vdop is not a scientific sound method for interpretation. Much more justification needs to be made.

➔ We improved all figures except Figure 1.

➔ We added the explanations for the thresholds of Doppler velocity in the draft. For drizzle, according to the Glossary of Meteorology of the American Meteorological Society, the diameter of a drizzle is less than 0.5 mm, and the terminal velocity is 2.068 m/s with 0.5 mm at the surface based on Foote and Toit 1969. Mosimann 1995 investigated the degree of snow crystal riming using vertical Doppler radar. He found that the degree of riming is proportional to the Doppler velocity and that there is a large fraction of graupel with the Doppler velocity greater than 2 m/s (fig. 3 in Mosimann 1995). In this classification, we did not consider the effect of air density. This classification has uncertainty from vertical air motion and air density. We think the impact of these two terms is not significant. This study does not aim for an accurate classification of hydrometeors but rather for a quantitative intercomparison of models on the same basis. We also think the name of categorization is not perfect, because the Doppler velocity has information on mixtures of different hydrometeors. The naming of this classification is related to the average Doppler velocity to understand the model's performance. Each model has their own categories of hydrometeors and characteristics like terminal velocity, density, and size distribution. The motivation of this study is that we need to compare hydrometeors with the same criterion as the

Doppler velocity. We want to use the same categorization for the understanding of microphysics schemes or intercomparisons of GSRMs. We added the paragraph in the introduction part: One of the motivations of this study is to evaluate and compare the vertical distribution of hydrometeors of GSRMs using the same observational criteria. According to Roh et al. 2021, the horizontal distribution of outgoing longwave radiation of GSRMs is similar, but the simulated vertical distributions of hydrometeors of GSRMs are very different in the intercomparison data (Stevens et al. 2019). Each model used its own assumptions about the size distribution and terminal velocity of hydrometeors. We believe that the Doppler velocity is one of the criteria for understanding and constraining the vertical distributions of hydrometeors between GSRMs using observations.

➔ We investigated the impact of vertical air motion on the Doppler velocity using our satellite simulator. We removed the vertical air velocity about the calculation of Doppler velocity (Fig. 2). When we removed the vertical air velocity, the results were consistent with the control results (Fig. 1). However, the frequencies were concentrated in NSW6, and there was no fraction of the upward category. Most of the differences are less than 2% in the classifications. We added Fig. 2 and explanations in the revised draft.

[Figure]

Figure 1: The categorizations of the hydrometeors in NICAM simulations for NSW6 (top) and NDW6 (bottom) in case 1 (left) and case 2 (right).

[Figure]

Figure 2: The same as Figure 1 but for only calculations of Doppler velocity without vertical air motion.

**Specific comments:**

Line 27: "... sampling of ...", this sentence needs to be reworded. You mean footprint of space-borne radar? Spatial sampling scales?

➔ It is different from the footprint. For example, the horizontal sampling of CPR is approximately 500m, and the footprint is approximately 800m (e.g. Kollias et al. 2014). I changed "the along-track sampling".

Line 34: "... in the same body ..." needs to be reworded. You mean same space craft?

➔ We changed it based on your comment.

Line 38: "synergetic"? Maybe using "synergistic"?

➔ We changed it based on your comment.

Line 90: I think the variable in Figure 1 is "precipitation rate".

➔ We changed it based on your comment.

Line 102 to 107: You did not mention how the simulations do for Case 2. The simulations missed the part of precipitation over the ocean to the east near 36 and 37 north latitudes. This should be added.

➔ We added the explanation about your notice about precipitation.

Fig. 2 caption: What is the unit of the color scale? Percentage? Please be specific. Also note this figure is for observation.

➔ We added the unit of the color scale.

Line 119: It should be Figure 2 c and d.

➔ We changed it based on your comment.

Line 120: Why "Doppler velocity is free to attenuation"?

➔ For the precipitation area, the observed radar reflectivity is not reliable because of the attenuation. The Doppler velocity is not attenuated, but the data quality is not good in the highly attenuated areas.

Line 123: "Two high-frequency modes are near the melting layer..." This description is rather vague. It is just the difference above and below the melting layer.

➔ We modified to "there are two different modes above and below the melting layer."

Line 126 "less than -2 m/s"? The figure shows "greater than -2 m/s". Do you mean the absolute value?

➔ It is not an absolute value. The rimmed ice particle has a Doppler velocity of less than -2m/s, like -4 or -5m/s.

Line 128: if you talk about aliasing, I think you should give the CPR's measurement range. When the velocity is out of the range, aliasing would happen.

➔ I explained the range of the Doppler velocity in the next sentence.

Line 129: Not using radar reflectivity is due to strong attenuation, right? You'd better mention this when you talk about the reflectivity attenuation earlier.

➔ We moved the sentence to the paragraph describing attenuation.

Last paragraph in Page 6: It looks like it is about Figure 3. Please refer to Figure 3.

➔ We referred to Figure 3 in the first sentence.

Comparing Fig. 3 and 2: why you use CFADs vs. joint histogram? Are they different in your paper? Give units to your plots.

➔ For understanding the vertical structure of the radar reflectivity and Doppler velocity, the CFADs are better because of the different sampling numbers per height. For the quantitative analysis, we thought the joint histogram was better than CFADs.

Comparing Fig. 3a to Fig. 2a: Are their differences solely due to unfolding? Why there is so much difference from 8 km and above?

➔ The difference is the normalization by each height or normalization by total height. The difference is the number of data samples at each height. The number of sampling data is not so

many in case 1 above 8 km. So, the distribution is different between the CFADs and the joint histogram.

Line 162: reword it to "produces a large bias and makes the results unreliable."

➔ We changed it based on your comment.

Line 164: Is it 2%? From the black solid line, I read 0.997, from the black dash and orange lines, I read 0.9985 to 0.999. So, it should be 0.3% or less. Why do you think it's 2%?

➔ We changed it by 0.2% based on your comment.

Line 164-165: are you talking about the simulated PDF of the Doppler velocity or the vertical air velocity?

➔ We talked about the simulated PDFs of the vertical air velocity.

Line 169: About the results are affected by the horizontal resolution of the model. The issue of the dependence of the vertical air motion on model resolution needs more details. Please restate the resolution of the model you are using and the coarse resolution you are referring to. Please also refer to previous studies about the dependence of vertical air motion on model horizontal resolution (e.g., Lebo and Morrison 2015 Monthly Weather Review page 4355-4375 or other study that you find appropriate.)

➔ Thank you for your interesting comments. We think the impact of horizontal resolution on the Doppler velocity and the contributions of vertical air velocity on the Doppler velocity is different between large-eddy simulation (LES) and cloud system–resolving model (CRM). When we reduce the horizontal resolution in our model (CRM), the contribution of vertical air velocity decreases (Fig. 3). We have also done LES with a different regional model and a different single-moment microphysics. We found that the LES with a 100-m resolution reproduced the fifth upward regime in the joint histogram well. We think the upward regime is mainly related to the turbulences in clouds (observation also shows a similar upward Doppler velocity related to turbulences). We will prepare a paper about this issue. The LES with a coarser resolution (250m

resolution) has a lower fraction of the upward regime.

[Figure]

Figure 3. The resolution dependency of the cumulative PDF of the absolute vertical air velocity for the sampling data with larger than -40 dBZ in NDW6 for case 1 (left) and case 2 (right).

Line 178: It is vague for rain vs cloud water vs drizzle. Please explicitly state how rain, cloud water, drizzle is separated in your categorization method. In Fig. 5 there is no cloud water or drizzle shown.

➔ The separation among cloud water, drizzle, and rain in this categorization method is the size of the liquid hydrometeors. The cloud water and drizzle have less than 0.5 mm diameter, and the terminal velocity is less than 2 m/s. In Fig. 5, the rain terminal velocity with less than 0.5 mm consists of the drizzle and cloud water categorization. However, two microphysics schemes have only a rain category in the scheme, which consists of rain and drizzle in this classification.

Figure 6: text on this figure is not readable. You need to adjust the location of the text for hydrometeor percentages, and the color of the text that labels the regions.

➔ We improved the figure based on your comment.

Line 193: You show the choice of microphysics scheme has more influence as compared to the case dependence. So, how this method could be used to provide useful information to the EarthCARE retrieval

methods?

➔ We think these results are useful for understanding the uncertainty of the retrieval method. The retrieval method has its own assumptions about size distributions and terminal velocity. We think these results show the importance of the microphysical setting in developing the retrieval method.

Line 200: The cloud ice has 0 m/s terminal velocity as shown in Fig. 5 in the single moment scheme. How do you interpret the growth of snow from cloud ice from Fig. 6?

➔ In Fig. 6, the growth of snow from cloud ice is not clear. The maximum and minimum Doppler velocity show the increase of Doppler velocity from the top to the melting layer. The main reason is the autoconversion process from cloud ice to snow occurs at higher than 12 km (Fig. 7).

Line 218: "It decreases the sampling of liquid hydrometeors and upward motion (Fig. 7)" – Do you mean the attenuation from the spaceborne CPR in the liquid hydrometeor layer is more severe?

➔ It is not related to attenuation. The sampling areas of the satellite related to ice hydrometeors increased than the ground observation from 12 km to 20 km. There are more fractions of ice hydrometeors with 8 km, and it decreases the fraction of liquid hydrometeors in the total fraction.

Line 222: "we found an increase ...." no text is illegible in your figures, same here in Fig. 7 and 8. You need to use a table to show the comparisons.

➔ We improved the figures and added the tables from the figures.

Line 310: correct "thee" to "three"

➔ We changed it based on your comment.

*References:*

Foote, G. B., & Du Toit, P. S.: *Terminal velocity of raindrops aloft. Journal of Applied Meteorology (1962-1982), 249-253.*1969.

Mosimann, L.: An improved method for determining the degree of snow crystal riming by vertical Doppler radar. *Atmos. Res., 37, 305–323, doi:10.1016/0169-8095(94)00050-N, 1995.*

*Lebo, Z. J., H. Morrison, 2015, Effects of Horizontal and Vertical Grid Spacing on Mixing in Simulated Squall Lines and Implications for Convective Strength and Structure. Monthly Weather Review, 4355-4375.*

Kollias, Pavlos, et al. "Evaluation of EarthCARE cloud profiling radar Doppler velocity measurements in particle sedimentation regimes." *Journal of Atmospheric and Oceanic Technology* 31.2 (2014): 366-386.

Roh, W., Satoh, M., & Hohenegger, C.: Intercomparison of cloud properties in DYAMOND simulations over the Atlantic Ocean. *Journal of the Meteorological Society of Japan. Ser. II*, *99*(6), 1439-1451, 2021.

**To Reviewer 3:**

Recommendation: Major revision

The authors present a study about the potential of EarthCARE for observing Doppler velocities. In general, the topic of the paper is interesting and relevant for ACP. I recommend the paper to be accepted after major revision.

**Major comments:**

Language: The manuscript lacks clarity due to language problems. I recommend that the authors give the paper to a native speaker to make sure they write what they intent to say. Also, I would recommend to guide the reader better to explain why a analysis was performed in a certain way. For example, in section 4 I would recommend to stress that you start with an idealized simulation without instrument effects like the Nyquist range or random errors and then make the simulation more realistic step for step. Also, it should be stressed that the hydrometeor classification is not supposed to be a universally applicable one (at least I hope this is that case) but is only used to allow for a better comparison between model and observations.

The authors study a case related to a tropical storm with potentially high reflectivities that might lead to multiple scattering, how would that impact the results? What about other sources for measurement errors? E.g. cloud inhomogeneity or pointing uncertainty?

→ The revised draft was checked by a researcher whose first language is English based on your comment.

→ We agree that the multiple scattering impact also affects the results related to heavy precipitation cases. We think we can clearly understand the uncertainty from the multiple scattering after the launch of the satellite. Our expectation is that the multiple scattering is not significant because of the 800 m footprint and circular polarization. We need to filter out the data related to multiple scatterings to get better results. In this paper, we assume we use calibrated Doppler velocity from the multiple scattering and the point uncertainty for evaluations of a global storm-resolving model. We will filter out the data related to multiple scattering in simulations with the same criterion as the retrieval algorithm in the future.

The cloud inhomogeneity is not important in this resolution with less than 500m. Now, we focus on the evaluations of a km scale global model. The purpose of this study is the application of the Doppler velocity for evaluations of modeling groups.

L 160: "We assumed the contribution of vertical air velocity to Doppler velocity is relatively smaller than the terminal velocity of hydrometeors" This is a strong assumption that needs to be supported. Alternative, the authors could remove convective data points using a filter like in e.g. Mosimann, 1995.

→ We checked the upward motion using the observation data. The frequency of the upward motion is very rare. The time interval of our observation data is less than a second, and we used one-minute integrated data for the analysis. So, we think convective data points are reduced by the integration. Additionally, we investigated the impact of vertical air motion on the Doppler velocity using our satellite simulator. We removed the vertical air velocity for the calculation of

Doppler velocity (Fig. 2). When we removed the vertical air velocity, the results were consistent with the control results (Fig. 1). However, the frequencies were concentrated, and there was no fraction of the upward category. Most of the difference is less than 2% in the diagrams. We added Fig. 2 and explanations in the revised draft.

[Figure]

Figure 1: The categorizations of the hydrometeors in NICAM simulations for NSW6 (top) and NDW6 (bottom) in case 1 (left) and case 2 (right).

[Figure]

Figure 2: The same as Figure 1 but for only calculations of Doppler velocity without vertical air motion.

L 260: Without averaging, the performance of the Doppler observations is quite bad for the high mode as can be seen in Fig. 11. Is that the main message the authors want to convey with this paper? Spatial averaging would improve the results, why was it not considered?

➔ We checked the resolution dependency of the simulation results using NICAM. We found the impact of resolution dependency is not larger than the choice of microphysics schemes. So, we expect the 10km integration data to be useful for the evaluation or intercomparison of GSRMs.

Did the authors correct for the effect of changing air density on hydrometeor sedimentation velocity? Wouldn't such a correction be necessary for a threshold based classification?

➔ For the Doppler velocity, we considered the effect of changing air density. However, we did not consider the classification method. We think the air density affects the classification of the hydrometeors. We think the impact is not significant for the classification. Our purpose is a simple evaluation method for intercomparison or evaluation of GSRMs.

**Specific comments:**

L 119: Fig 1 -> Fig 2c-d

➔ We modified it based on your comment.

L 166; "When we use the threshold of 2 m/s for categorising the hydrometeors, 0.2 m/s of vertical air velocity affects the 10% bias". Does that mean that the authors expect that 0.2 m/s vertical air motion lead to a 10% error of the classification? This is only true if the data points would be equally distributed with Doppler velocity which is not the case.

➔ We removed the explanation about that.

L 239: Specify latitudes for low and high modes.

➔ We specify the latitudes like "The high and low modes will be used in the operation: low mode at latitudes of 60°–90° and high mode (−1 to 20 km) at latitudes of 0°–60° (Hagihara et al., 2022)."

Figs 1-3, 6-11. Please add labels to all x axis, y axis and colorbars

➔ We added the labels to all x-axis, y-axis, and color bars.

6: Why is there no data above 12 km?

➔ The vertical range of the observation data is until 12 km.

8: Not referenced in the text

Data availability: Where are the used simulations and radar observations available?

➔ Simulation data will be available, but we need to discuss opening our observation data to the public. We will upload the available data before publication.

Reference

Mosimann, L., 1995: An improved method for determining the degree of snow crystal riming by vertical Doppler radar. Atmos. Res., 37, 305–323, doi:10.1016/0169-8095(94)00050-N.

Woosub Roh
Atmosphere and Ocean Research Institute, The University of Tokyo
5-1-5, Kashiwanoha, Kashiwa-shi, Chiba, Japan
Phone No: 81- 04-7136-4371
Fax No: 81- 04-7136-4375
Email Address: ws-roh@aori.u-tokyo.ac.jp

---

## Author Response (AR2)

We appreciate the reviewers' constructive comments. We have revised our manuscript accordingly, and we hope the reviewer will find the revisions satisfactory.

**Review of "An evaluation of microphysics in a numerical model using Doppler velocity measured by ground-based radar for application to the EarthCARE satellite" by Roh et al., submitted to Atmospheric Measurement Techniques (AMT)**

[Article#: amt-2023-1997]

**To Reviewer 1:**

SUMMARY AND OVERALL ASSESSMENT

The manuscript deals with several aspects of how new EarthCare Doppler velocity (hereafter DV) measurements might address problems of hydrometeor identification using a variety of observations and simulations.

The revisions to the manuscript improved both the grammatical usage and presentation and methodology.

MAJOR COMMENTS

1. I appreciate that you added some context to specific values associated with the hydrometeor classification scheme chosen.

You state in your response: "This study does not aim for an accurate classification of hydrometeors but rather for a quantitative intercomparison of models on the same basis," and also "We think the accuracy of the classification's names is not very important in this study."

This, and the new text you added, suggests that the thesis of the manuscript is that the Doppler velocity vs. height space that you use throughout the paper (I will call this DV-H space) is useful for comparing model vs. model and model vs. observation. I agree with that assessment.

But the fact remains that you do break down this space into classes, and then use those classes to interpret the microphysics. For example, in Line 222 "The fraction of CW/drizzle is underestimated in both simulations". Strictly speaking, I can only accept that statement if I also accept that your CW/drizzle classification is correct. (In this case, I believe it's probably pretty good).

It is not unreasonable to ask whether these hydrometeor classifications, at least in the models, actually populate the regions of the DV-H diagram that you assert in the classification scheme imposed. You

provide some secondary evidence suggestion they do, for example in Figure 5 where you show the separation between different hydrometeor types. One way to demonstrate this would be to produce a version of one of the panels of Figure 6 (for example) where only one hydrometeor class at a time is shown.

→ The two microphysics schemes do not have a class called drizzle, but we think that drizzle could be understood as rain with a small size. A notable issue arises with the NSW6 scheme, which struggles to accurately depict small-sized rain, often classified as drizzle in the D-H observations. Despite the absence of a direct 'drizzle' classification in the model, recognizing this size-based categorization is crucial for understanding raindrop size distribution. This distribution, along with the terminal velocity of raindrops, significantly influences the evaporation process within the boundary layer. In our future research, we aim to conduct long-term simulations to compare models that effectively represent drizzle against those that do not, to further investigate these dynamics.

2. You should address how the integration time affects the implications of this analysis when your method is used on real EarthCare data. You show results collected over a geographic region during a 24 hour period. EarthCARE will have flown a over these storms in seconds, with a very small footprint, and likely not sampled them again.

→ Thank you for your comment. When we use satellite data with active sensors, we need to collect data for a more extended period (more than one month) and a large domain for statistical analysis. We have experience in the evaluation using satellite data like TRMM or CloudSat (Roh et al., 2014; Roh et al., 2017). When we used satellite data, we collected the data for more than one month and a large domain for statistical analysis. According to our experience, the characteristics of simulation data like CFADs were controlled by the microphysics schemes rather than the integration time.

MINOR COMMENTS

- Figure 7 was a welcome addition to show the overall small effects of vertical air motion. It's much more compelling than Figure 4, which unsurprisingly shows that velocities on the order of cm/s are much more common than velocities of m/s. The problem there is that certain interesting phenomena (heavy rain, buoyant updrafts, etc.) are crammed into that 0.2% that have absolute vertical velocities above 0.2 m/s. Also your analysis is presumably affected by the ~km scale width of a grid box.

→ Thank you for your comment.

- HG-SPIDER is indeed polarimetric, which could help with hydrometeor classification in the observations. However I accept this might be difficult to do in practice.

→ I am sorry for my mistake. I misunderstood the coauthor's comment. The HG-SPIDER was developed as a polarimetric radar. HG-SPIDER did not observe the depolarization ratio in these cases.

- Line 89 suggests that the Case 1 integration time is 24 hours, but Line 101 suggests it is 12 hours. Please clarify.

→ The integration time is 24 hours for simulations, but observation data is only available for 12 hours because of the instrument issues.

- Line 202: Doesn't NDW6 have the clearer separation? Or am I misinterpreting overlapping lines?

→ NDW6 shows the overlapping terminal velocity with 2m/s between graupel and large cloud ice or snow.

- Line 230: "We checked the impact"...

→ We changed it based on your comment.

- Line 234: Convection is often its own plural in this context (without the 's')

→ We changed it based on your comment.

- Line 253: "We focused the data"; perhaps replace with "Therefore we analyzed data"

→ We changed it based on your comment.

- Line 270: "The Joint-Simulator can simulate the EarthCARE CPR", removing "the signals like"

→ We changed it based on your comment.

- Line 338: "In this study we developed a methodology"...

→ We changed it based on your comment.

References

Roh, W. and Satoh, M.: Evaluation of precipitating hydrometeor parameterizations in a single-moment bulk microphysics scheme for deep convective systems over the tropical central Pacific, J. Atmos. Sci., 71, 2654–2673, https://doi.org/10.1175/JAS-D-13-0252.1, 2014.

Roh, W., Satoh, M., Nasuno, T.: Improvement of a cloud microphysics scheme for a global nonhydrostatic model using TRMM and a satellite simulator, J. Atmos. Sci., 74, 167–184, https://doi.org/10.1175/JAS-D-16-0027.1, 2017.

**To Reviewer 2:**

The revision improved the presentation of the work and made it easier to read. However, it is still recommended that the authors continue to improve some confusion parts in the manuscript and its overall readability before it is accepted.

The model information of the terminal velocity for different hydrometeor categories is only used to justify the thresholds that are used to set categories in the observations and the simulated Doppler velocity. Although the terminal velocity information could be better used in the analysis, due to the scope of this study, it is ok.

→ Thank you very much for your feedback. Based on the comments received from both the editor and the reviewer, I have made further refinements to the manuscript to enhance its clarity and readability. I appreciate the suggestion to utilize the terminal velocity information in our analysis more effectively. While the current scope of our study primarily leverages this data to justify observation and simulation categorizations, I acknowledge the potential for a deeper exploration of this aspect in future work. We remain committed to addressing any remaining areas of confusion and improving the manuscript's overall coherence in preparation for acceptance

Woosub Roh
Atmosphere and Ocean Research Institute, The University of Tokyo
5-1-5, Kashiwanoha, Kashiwa-shi, Chiba, Japan
Phone No: 81- 04-7136-4371
Fax No: 81- 04-7136-4375
Email Address: ws-roh@aori.u-tokyo.ac.jp